# Adapting Digital Social Prescribing for Suicide Bereavement Support: The Findings of a Consultation Exercise to Explore the Acceptability of Implementing Digital Social Prescribing within an Existing Postvention Service

**DOI:** 10.3390/ijerph16224561

**Published:** 2019-11-18

**Authors:** Karen Galway, Trisha Forbes, Sharon Mallon, Olinda Santin, Paul Best, Jennifer Neff, Gerry Leavey, Alexandra Pitman

**Affiliations:** 1School of Nursing and Midwifery, Centre for Evidence and Social Innovation, Queen’s University Belfast, Belfast BT7 1NN, UK; T.Forbes@qub.ac.uk (T.F.); O.Santin@qub.ac.uk (O.S.); 2Faculty of Wellbeing, Education and Language Studies, Open University, Walton Hall MK7 6AA, UK; Sharon.mallon@open.ac.uk; 3School of Social Sciences, Education and Social Work, Centre for Evidence and Social Innovation, Queen’s University Belfast, Belfast BT7 1NN, UK; P.Best@qub.ac.uk; 4Elemental Software, Derry (HQ), Derry BT48 6HP, UK; jennifer@elementalsoftware.co; 5Bamford Centre for Mental Health, Psychology Research Institute, Ulster University, Newtownabbey BT37 0QB, UK; Gerry.Leavey@Ulster.ac.uk; 6UCL Division of Psychiatry, University College London, London WC1E 6BT, UK; a.pitman@ucl.ac.uk

**Keywords:** social prescribing, suicide bereavement, postvention, co-design, complex intervention development, implementation science

## Abstract

This paper describes a consultation exercise to explore the acceptability of adapting digital social prescribing (DSP) for suicide bereavement support. Bereavement by suicide increases the risk of suicide and mental health issues. Social prescribing improves connectedness and empowerment and can provide digital outcomes-based reporting to improve the capacity for measuring the effectiveness of interventions. Our aim was to consult on the acceptability and potential value of DSP for addressing the complexities of suicide bereavement support. Our approach was underpinned by implementation science and a co-design ethos. We reviewed the literature and delivered DSP demonstrations as part of our engagement process with commissioners and service providers (*marrying evidence and context*) and identified key roles for stakeholders (*facilitation*). Stakeholders contributed to a co-designed workshop to establish consensus on the challenges of providing postvention support. We present findings on eight priority challenges, as well as roles and outcomes for testing the feasibility of DSP for support after suicide. There was a consensus that DSP could potentially improve access, reach, and monitoring of care and support. Stakeholders also recognised the potential for DSP to contribute substantially to the evidence base for postvention support. In conclusion, the consultation exercise identified challenges to facilitating DSP for support after suicide and parameters for feasibility testing to progress to the evaluation of this innovative approach to postvention.

## 1. Introduction

The aims of this work were to (1) consult on the potential value of social prescribing for addressing the complexities of suicide bereavement support, (2) to address gaps in the evidence-base for interventions supporting people bereaved by suicide, and (3) explore how digital solutions could help overcome these challenges. Social prescribing, also known as community referral, provides a means of enabling healthcare professionals to refer people to community-based, non-medical support [1].

Those who experience suicide loss are at a higher risk of suicide; therefore, suicide bereavement support is crucial to suicide prevention, featuring prominently in most suicide prevention strategies. Accessing support can be difficult due to self-stigma, public stigma, and complex family situations, and stronger evidence is needed to understand what works, for whom, and in what circumstances [2].

One of the well-established risk factors for suicide and suicidal behaviour is a family history of suicide [3], but beyond genetic risk, more recent research has shown that spouses [4] and unrelated peers [5] are also at risk of suicide and suicide attempt after suicide bereavement. The number of people affected by a single suicide loss is estimated at 60 friends, relatives, and colleagues [6], but with estimates as high as 135 people in the wider social network [7]. There is evidence that partners and ex-partners, parents, and in particular mothers of adult children who die by suicide, have significantly elevated risks of suicidal behaviour and death by suicide [8,9]. Increased risk of suicide in other kinships is less well researched; however, in one UK study, approximately 50% of next of kin, including partners, parents, siblings, and offspring, reported feeling suicidal after losing a family member to suicide [10]. It is likely, but not always the case that the individuals most closely related to the deceased are most adversely affected by the death [11]. The age-standardised suicide rate in the UK is 11.2 deaths per 100,000 population, representing 6507 suicide deaths in 2018 [12]. Given the above estimates of the number of people affected by each suicide, between 390,400 and 878,400 may be bereaved by suicide every year in the UK. This group are, therefore, potentially at increased risk of mental health difficulties, including non-fatal and fatal suicidal behaviour [8,13].

People bereaved by suicide commonly experience increased anxiety, depression, post-traumatic stress disorder (PTSD) [4], prolonged grief, impaired social functioning, and increased suicide risk [8,13,14,15]. Along with a sense of loss and isolation, self-stigma can impede help-seeking [16,17] reducing access to protective factors, including professional and informal support networks [18,19,20]. Beyond these barriers, the bereaved may also face complex family dynamics exacerbated by the circumstances surrounding a suicide, which can prevent or delay people seeking support [20]. The economic impact of a single suicide has been estimated at approximately £1.67 m per death (using 2009 prices), with costs associated with each suicide bereavement estimated at almost £50,000 [21]. The magnitude of human and societal costs presents a convincing public health argument for improving support for people bereaved by suicide. Currently, the existing evidence suggests that a mixture of therapeutic, social, and educational interventions may be required [2].

Social prescribing (SP) is a model of support that connects people to a range of non-medical community-based services via a link worker. A link worker is an individual who facilitates co-creation of a social prescription, usually by combining education, information, and shared decision making. Social prescribing schemes tend to link people to support for physical and mental health as well as providing social, educational, housing, legal, and other types of support. Emerging evidence suggests that SP can improve connectedness, increase sense of empowerment, and reduce health service use [22,23]. Importantly, in the UK, SP has been promoted as a key area for service development, with NHS England pledging to train 1000 social prescribing link workers by 2021 [24]. This is supported by emerging, promising evidence on the acceptability enjoyed by SP for non-NHS community-based support, and on reducing the reliance on NHS healthcare services to address the social need [23].

Increasingly, organisations tackling health and wellbeing issues faced by communities are considering the value of digital SP (DSP) platforms as part of the effort to increase access to services that reduce health risks. These DSP software platforms provide the intelligence to manage an SP network of all of the services, users, and professionals who need to be involved. This helps with uptake, security, integration with existing processes, and makes life easier for the GP or the person making the initial referral, easier for the link worker who receives the referral, and more likely that the individual seeking services may get benefit from receiving an SP.

Much further work is required to embed DSP solutions into organisations across health, housing, local government, and the voluntary, community, and social enterprise (VCSE) sector to support them to deliver SP and analyse the data generated. Much more work is also required in terms of information systems’ integration, and this is very much the direction of travel across the SP landscape. Indeed, one of the main opportunities digital options afford, but equally one of the main challenges is the connecting of information between different organisations and connecting different systems to enable those involved in SP to best serve people and communities.

The aims of this work were constructed to explore the challenges of testing SP in a regional system providing support after suicide in Northern Ireland, a devolved nation of the UK. In Northern Ireland, police officers report suspected suicides at the scene by completing a ‘Sudden Death’ form (as part of the bereavement support system, known as the SD1 process). This offers the bereaved an opportunity to consent to bereavement support. The SD1 process can also trigger a Community Response Plan (CRP) to provide a greater level of support where needed [25]. Approximately 50% of next of kin provide consent to bereavement support. Their details are passed to public health bodies, including the NHS and voluntary and community sector (VCS) providers, commissioned to provide counselling, alternative therapies, and other support services. Evaluating the impact of the current service provision has been challenging due to the complexities of monitoring this level of inter-agency involvement [25]. Therefore, the research team sought to explore the acceptability of layering a digital SP (DSP) system into existing service configuration, in order to address three priorities identified in recent work including the views of people with lived experience of suicide bereavement, about support services [25]. These recommendations, supported by the wider literature, were; (1) to integrate the support pathways and information systems between agencies, (2) to open up referral routes into a wider range of suicide bereavement support services, and (3) to measure and record outcomes to address the need for a stronger evidence base for suicide bereavement support [25,26].

## 2. Materials and Methods

The four-step methodological approach was informed by; the Medical Research Council (MRC) complex intervention development guidance [27], the concept of co-design [28], and the potential of implementation science [29].

Input from the MRC intervention development guidance was limited to stage one; assessing contextual factors for modelling processes and outcomes [27]. Co-design guidance was sought from INVOLVE [30], the New Economics Foundation (NEF), and from applied co-design modelling [28], while the theoretical PARiHS (Promoting Action on Research Implementation in Health Services) framework (Rycroft-Malone, 2008) guided input from the discipline of implementation science. The PARiHS framework proposes that successful implementation is represented as a function of the nature and type of evidence, the qualities of the context in which the evidence is being introduced, and the way that the process is facilitated [31].

Step one: *contextual analyses* and *asset mapping* were carried out to identify partners involved in the existing system of support for people bereaved by suicide. This was to allow the researchers to marry academic evidence with ‘real-world’ expertise of service providers and to facilitate introducing the concept of SP and a demonstration of the DSP platform. The process involved an initial roundtable discussion (*n* = 10) held on 27 April 2018 at Queen’s University Belfast, to identify potential partners (service providers) followed by face-to-face meetings with providers and commissioners (*n* = 16 organisations). Visits to organisations took place from 27 April to 15 June 2018. The visits provided knowledge on the *context* into which the evidence will be generated, and potentially introduced and supported developing new relationships [32]. Handwritten notes were kept of discussion points at each meeting. All partners identified during this process received an invitation to a co-design workshop to further explore the mechanisms of DSP to develop a shared understanding and to explore acceptability.

Step two: prior to the co-design workshop, we invited delegates to contribute to the design of the workshop via an online consultation tool, Well Sorted [33]. Well Sorted is an online sorting tool for inviting and grouping contributions from stakeholders in advance of meetings or further discussion. It has been created by a research group Texture Lab, based in Edinburgh, Scotland, and is free to use. Invitees were asked to suggest up to four priorities and/or challenges in providing suicide bereavement support.

Step three: we welcomed delegates to participate in a co-design workshop held on 20th June 2018 at a conference centre near Belfast (*n* = 29; 19 females and 10 males). Delegates represented the statutory sector (*n* = 9), voluntary, community, and social enterprise (VCSE) sector (*n* = 8), higher education institutions (*n* = 9), and commercial/independent consultancies (*n* = 3). Organisations included the Public Health Agency and the Department of Health; the Health and Social Care Trusts, the Police Service of Northern Ireland (PSNI); four Higher Education Institutions; Elemental Software (providers of digital social prescribing platform), Barnardo’s (children’s charity), New Life Counselling (Counselling and peer support), Family Voices Forum, and Survivors of Suicide (peer-led advocacy groups). The co-design workshop began with an overview of DSP and the rationale for testing it on people bereaved by suicide. Group discussions were designed to consider the priorities for supporting people bereaved by suicide in the context of DSP and to gather details of client pathways and roles in the support journey. Group composition combined a range of input from sectors to capture variety in expertise and perspectives. Research team members facilitated discussions and took notes. During the first part of the workshop, four groups were asked to discuss two topics each (*n* = 8 topics). These were priorities and/or challenges identified in Step 2. A facilitator for each group captured a summary of discussions on flip chart pages. These included key challenges of providing suicide bereavement support and, where possible, potential solutions where these could be identified. In the second part of the workshop, the discussion focused on client pathways, including service roles and responsibilities. This was discussed in the context of how best to introduce and implement DSP. For example, one key question was; “Who would be best placed to perform the role of ‘link worker/navigator’?” Details were documented using a mixture of field notes and flip chart pages, including technical and practical parameters to support a feasibility study of social prescribing for suicide bereavement. Based on the notes made during these workshop discussions, two researchers (KG and TF) produced a written summary of the topics discussed. This draft was emailed to group facilitators, incorporating their comments into successive versions. This exercise was explicitly not a formal qualitative analysis but the documentation and synthesis of a range of viewpoints. The findings of this analysis were circulated by email to all delegates, explaining that the findings would be used to support a funding application and also a peer-reviewed publication.

Step four: we gathered workshop delegate views on the sort of outcomes that they would want to see measured, in a feasibility test of DSP for suicide bereavement support. This was driven by the need for addressing gaps in the evidence-base for interventions supporting people bereaved by suicide. Well Sorted [33] was used (post-workshop) to specifically seek delegate views on the most important outcomes that should be measured. See Figure 1 for an overview of the methodological steps.

### Statement on Ethics 

The activities described in this paper were conceptually designed, delivered, and reported as a co-design consultation process, and therefore do not meet the requirements for research ethics approval processes, according to the UK’s Health Research Authority [34]. All delegates were fully informed of the aims of the project and the intended use of information shared. All communications, recording of information, and other processes were administered according to the ethical principles outlined in the Declaration of Helsinki.

## 3. Results

We contacted a total of 30 organisations across Northern Ireland (including 7 statutory organisations and 15 voluntary and community sector organisations) that provide support to people bereaved by suicide. Our engagement rate was high, with 74% of organisations contacted agreeing to take part in the workshop (100% of statutory organisations *n* = 7/7 and 66% of voluntary and community sector organisations *n* = 8/12]. Across approximately 32 delegates registered for the workshop, we received 40 suggestions in the pre-workshop consultation (from 17 delegates), and 18 responses to the post-workshop follow up (from 8 delegates).

### 3.1. Step One

In step one, as described above, we conducted contextual analyses and context mapping through a round table discussion with 10 service providers and visits to providers and commissioners at 16 organisations. This step provided an opportunity to gauge the broad scope of suicide bereavement service provision offered across Northern Ireland and to assess receptiveness of service providers to the concept of adapting DSP for suicide bereavement support. This contextual readiness is key to successful implementation [35], and in this context, step one provided an opportunity to develop relationships and partnerships; therefore, it enhanced engagement rates and commitment to the workshop processes.

### 3.2. Step Two

In step two, we used the online consultation tool to identify the following eight key topics: the stigma of suicide bereavement, reluctance to access support (men and nihilism), suicide and substance misuse, the impact on wider communities, supporting beyond next of kin, matching type of support to need, barriers to implementing digital systems, and timing of offering support.

### 3.3. Step Three

In step three, each topic was discussed in small groups at the one day workshop. For each topic, we identified and considered three challenges and three potential solutions involving DSP or by any other means. A summary of these discussions is given below and in Table 1.

#### 3.3.1. Addressing the Stigma of Suicide Bereavement and Impact on Help-Seeking

Delegates reported on common grief responses, such as guilt and shame, felt by the suicide-bereaved, which require sensitive and responsible discussion. It was felt that suicide awareness education should focus on normalising the word suicide but not the behaviour. In summary, delegates agreed that DSP could support overcoming the stigma of suicide bereavement by providing sensitive information, a sense of connectivity, and safe, knowledge-based support that is person-centred. Thorough monitoring was recognised as the key to measuring any reduction in stigma.

#### 3.3.2. Reluctance to Access Support

One of the primary perceived challenges of providing postvention support is not knowing how many people have been bereaved by any particular suicide death and how to provide access to this wider population at risk. The next of kin is the primary contact point, and where no consent is provided, support is not offered. It was suggested that those initially resistant to support could perhaps be re-contacted at a less acutely distressing time. Help-seeking should, therefore, be made more accessible through outreach to those bereaved as early as possible in order to provide a conduit to others and to provide psychoeducational reassurance around perceived vulnerabilities and how to tackle them.

#### 3.3.3. Was It Accidental Death due to Drugs/Alcohol or Was It Suicide?

Confirming cause of death and the role of drugs or alcohol in suicide raises sensitivities regarding the term ‘suicide’. The term ‘traumatic death’ may be more appropriate where coronial processes are incomplete or delayed. There were also concerns raised at the workshop about the community reactions to ‘speculative’ or ‘grey’ suicides. It was agreed that postvention outreach requires careful consideration of language.

#### 3.3.4. The Impact on Wider Communities

The context for this discussion included schools, colleges, universities, work places, and minority community groups. Delegates emphasised the challenges involved in protecting the confidentiality of the deceased individual, especially when whole communities are impacted. A distinction was made between interventions that carried a risk of heightening emotions versus raising awareness, and community response plans were also discussed. In order to counter any negative impact of postvention work, it was felt that there needs to be wider roll-out and publicity of support services. Future development of DSP was seen as a means of providing a self-referral option to postvention services, to allow those who need support to come forward independently, in response to carefully marketed public health postvention provision.

#### 3.3.5. Supporting Those Beyond the Next of Kin

The primary difficulty in accessing those beyond next of kin was perceived to be the issue of seeking consent to offer support. This consent is sought by police officers at the scene of the death. In the case of a bereaved wife, it might be expected that she would have the same GP as her husband, and thus be easily identifiable. The process for gaining her consent to be contacted by postvention support services would, therefore, be relatively feasible. The same might not apply to adult children, cousins, colleagues, and others affected emotionally by the loss, who might not have the same GP, and therefore would be less identifiable. Self-referral options were therefore considered essential to reach those most affected, who may not be those listed as next of kin. It was suggested that the support service(s) could be publicised via GPs, Schools, Healthy Living Centres, Health Trusts, and the VCSE sector. Overall it was agreed that we need to develop more effective means of identifying and supporting those who are affected beyond the next of kin. A self-referral option is a desirable aim.

#### 3.3.6. Matching Types of Support to Needs

Addressing and catering for the range of individual needs was identified as a major challenge, particularly in relation to interventions addressing emotional needs tailored to be age-appropriate, gender-appropriate, and socially/culturally appropriate, and in providing equity of access in rural and urban settings. It was suggested that gaps in provision should be identified and existing supports and systems could help with this process, for example, the Coroner’s Liaison Officers (who currently provide support for the inquest process). Delegates were enthusiastic about the potential of DSP to provide a wider range of person-centred responses to bereavement support needs.

#### 3.3.7. Does the Term Digital Put People off?

The discussion raised practical questions as to whether providers would be able to operate a digital SP platform in situ and whether it would create or fuel any inequalities or social disadvantages. Aside from recognising the challenge of limited internet access in rural areas, suggestions included capacity building through software training and use of a champions models (training for individuals to become ‘champions’ to disseminate knowledge within their workplace environment) to help drive changes in services.

#### 3.3.8. Timing of Service Provision (in Relation to Loss)

The final topic for discussion focused on the timing of support in relation to the bereavement. It was suggested that follow-up could be repeated at various stages of post bereavement. It was generally agreed that counselling should not always be offered in the early stages of bereavement and that support should be tailored to the needs of the individual. For example, to include practical, legal and financial affairs, and information about the Coroners’ inquest processes. Regarding timing, additional training for police was seen as an important way to enhance access to support. Discussions in the workshop concluded that those bereaved should be offered appropriate support at various stages post bereavement, recognising that initial practical support followed by emotional support is often appropriate.

The second half of the workshop focused on service delivery mechanisms, including roles and responsibilities within the SD1 postvention service, to address the methodological challenges of testing DSP. In NI, the support system known as the SD1 Process, is relatively unique because it is an outreach public health service [25]. The model we discussed in this workshop involved referring an individual to a link worker, who would meet with the bereaved person to assess a range of needs and use the digital SP platform to select or ‘prescribe’ acceptable community-based non-medical interventions, which may include social activities as well as individual advice and support, for example, counselling. Creating the prescription together would provide an opportunity for the bereaved person to identify and address their most pressing needs, and for the prescriber to probe as to whether suggestions were well-received.

The discussions captured the various pathways to suicide bereavement support that exist in each of the five Health Trusts in NI. Figure 2 illustrates this bereavement support process, staged by the gatekeeper, commissioner, and support roles. The ways in which DSP could potentially enhance the support services are listed beneath the role descriptions. At the gatekeeper stage, DSP could broaden the referral pathways to a link worker via primary and secondary care and also by self-referral. For commissioners, a DSP platform would allow for consistent monitoring of referrals, uptake, assessments, and outcomes. Similarly, for link workers, the DSP could provide a single point of reference for connecting people to a wider range of participating services and for completing electronic assessments and outcome measures.

### 3.4. Step Four

The findings of step four, a post-workshop consultation via email using the Well Sorted tool [33], indicated the main outcomes of a feasibility study that would be of most interest to the expert stakeholder group. The responses reflected a clear priority to capture the impact of DSP on individual client mental health and grief experiences. The feedback also indicated the importance of a measure of any improvement in awareness of support and level of support received, a measure of clients’ sense of personal empowerment, an indication of the level of engagement with support and continuity of support pathway, and a measure of whether or not practical support needs are met. Service-level and economic outcomes were also featured in the consultation feedback; these included any increased access resulting from introducing DSP to allow commissioners and providers to assess capacity pressures, as well as any wider impact of DSP on service providers to measure and assess the economic impact of DSP.

## 4. Discussion

This paper provides a unique insight into the steps we used to develop a co-design regional postvention suicide bereavement support service that could embrace DSP. The co-design approach brought together expert views on the priorities to be addressed for developing a much-needed evidence base for existing suicide bereavement support services. A positive consensus emerged that the introduction of a DSP system could add considerable value to existing systems and could be used to enhance services in a variety of important ways including; reducing the stigma of help-seeking, providing a wider reach to offer improved access to services, and improving the monitoring and measurement of impact. Importantly, testing the concept of DSP answers calls to enhance the evidence base for postvention support [26] whilst harnessing newer technologies.

Of particular note is the value of our methodology in establishing the contextual readiness of stakeholders to engage in future work as required to develop this program of research, in order to test for evidence of effectiveness. Using a combination of *contextual analyses* and *asset mapping*, followed by pre- and post-workshop consultations, we confirmed the contextual readiness of key stakeholders to introduce SP to the postvention service and contribute to its evaluation. In consultation with stakeholders about day-to-day challenges and successes, we tapped into the existing expertise to consult on the service structures context, providing insights on the facilitation of feasibility testing. Our process follows principles of best practice for establishing SP systems, by employing an implementation approach, developing shared understanding and new relationships, and assessing organisational readiness and of leadership and organisational factors [32].

### 4.1. Ethical Challenges Identified

Our discussions highlighted that some individuals are reluctant to access support and that some demographic groups may find DSP unappealing. We were clear that implementing DSP should not compromise the support available to those who would not want to engage with DSP. Issues of confidentiality were a major concern in relation to referral processes. Police attending an apparent suicide would need to be mindful that a bereaved person might not want to be identified as someone in need of suicide bereavement support. Similarly, stakeholders discussed systems for contacting bereaved individuals in the network and how to do this without compromising confidentiality. The ethical challenges of this were such that a reliance on self-referral was felt to be the only solution for those beyond the next-of-kin.

### 4.2. Limitations

This paper describes the results of a workshop consultation for intervention development, to prepare for feasibility testing, and is, therefore, reporting on expert opinion as opposed to results of formal research. While the concept and potential of DSP were enthusiastically embraced, workshop delegates acknowledged the challenges of negotiating multiple information technology infrastructures, particularly the integration of new platforms within NHS settings.

## 5. Conclusions

Designing and monitoring a system to provide support after suicide remains a complex challenge. Our experience of consulting with stakeholders over the acceptability of adapting digital social prescribing (DSP) for suicide bereavement support, and then using their expertise to co-design such a system in Northern Ireland, was very positive. We gained valuable insights from the commissioners and providers of an existing inter-agency postvention bereavement support service operating on a regional level in Northern Ireland. The experts we consulted felt that social prescribing and, in particular, digital social prescribing has the potential to overcome some of the barriers and challenges the current service faces in providing postvention support on a regional basis. Specifically, it has the potential to connect more people to support and to incorporate measurement of individual and service-level outcomes. Based on this evidence of acceptability and valuable discussions about feasibility, the next stage in this process would be to conduct feasibility testing and cost/outcome measurement as a prelude to a full cost-effectiveness evaluation.

## Figures and Tables

**Figure 1 ijerph-16-04561-f001:**
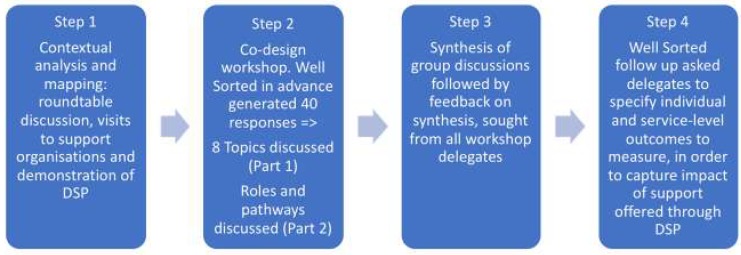
Methodological steps describing the co-design of a digital social prescribing intervention development workshop.

**Figure 2 ijerph-16-04561-f002:**
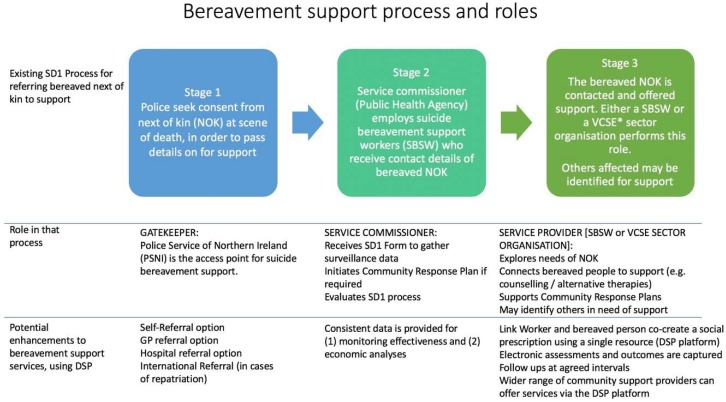
Representation of workshop discussion (Part 2) focused on client pathways and support roles, including the potential of digital social prescribing (DSP). * VCSE Voluntary Community and Social Enterprise sector.

**Table 1 ijerph-16-04561-t001:** Themes and priorities for enhancing postvention support after suicide identified in Step 2 and discussed in Step 3.

Priority Identified	Discussions and Potential Solutions
The stigma of suicide bereavement	DSP offers a potential solution for overcoming the stigma of suicide bereavement by providing connectivity and a safe, knowledge-based support system through personal empowerment via the range of support that could be offered.
Reluctance to access support	Educating people about vulnerabilities and how to address them could ease the stigma around seeking support.
Was it accidental death due to drugs/alcohol or was it suicide?	The sensitivities around suicide deaths involving drugs and/or alcohol misuse, and also acceptance of the term ‘suicide’, requires language consideration within any system of support. ‘Traumatic death’ is more sensitive than ‘suicide’.
The impact on wider communities	Distinguish between post-suicide activities that carry a risk of leading to heightened emotions versus those that are useful for raising awareness. To counter any negative impact, there needs to be wider roll-out and publicity of support services. Future development of the digital solution could also include self-referral.
Supporting those beyond next of kin	There is a need to develop more effective means of identifying and supporting those beyond the next of kin. See also: Reluctance to Access Support and Impact on Wider Communities.
Matching types of support to needs	Gaps in provision should be identified. Consider links with Coroners’ services.
Does the term ‘digital’ put people off?	Capacity building for using a digital social prescribing platform should be offered through software training. Use of *champions models* to cascade awareness and training could assist with changing the management process.
Timing of service provision (in relation to loss)	The bereaved clients could be followed up at different stages post bereavement; support should be tailored to the needs of the individual; initially focusing on practical hurdles, later providing emotional support options.

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
