# Peer review of "Adapting Digital Social Prescribing for Suicide Bereavement Support: The Findings of a Consultation Exercise to Explore the Acceptability of Implementing Digital Social Prescribing within an Existing Postvention Service"

_ijerph, 2019, doi:10.3390/ijerph16224561_

Round 1

Reviewer 1 Report

The title of this paper was immediately appealing as it offered up both the promise of a co-designed intervention and digital social prescription adapted for a suicide postvention service. However, as the paper progressed I was disappointed by my misapprehensions. I had anticipated a co-design study involving the design and possible implementation of digital social prescription for postvention support but found 'the results of an expert consultation for intervention development, to prepare for feasibility testing' (pg 8).  I think it is fair to say that whilst the expert consultation activities and processes described in the paper may be significant in terms of implementation science they do not qualify as co-design - and bear little similarity to the methodology/methods in the Santin et al paper that is cited as the source (and a co-author on the paper). I would recommend that the authors realign the focus of the paper to more accurately frame the activities undertaken and clarify the contribution of the paper. It does seem as though the paper fits more appropriately with implementation science - as it seems as though the intention is to implement evidence and practice around digital social prescribing into the postvention space through stakeholder engagement.

I offer the following comments on constructive ways I think the authors could enhance the paper and clarify the contribution of their work to the literature. 

Introduction

The substantive focus of the paper is digital social prescription however, the majority of the introduction is dedicated to examining the significance of postvention support in terms of population health re risk for suicide, impact of bereavement from suicide and economic impacts. When the paper finally gets to digital social prescription in the fourth paragraph the reader is only given 2 lines. I was left very unclear as to what social prescribing is, where it has come from, why the language of prescription is being used and how it works in practice. In addition to a fuller conceptual discussion, the paper requires a more thorough review of the literature in this field and connections established with some of the key claims - for example, the authors state that self stigma is a barrier to accessing support and that social prescribing can increase sense of empowerment - but there is no explanation as to how - or indeed what is meant by empowerment (empowered to access services?). There is also a confusing claim that whilst social prescribing connects people to services it can 'reduce service use'. The authors also suggest that digital SP will be able to 'measure and record outcomes' but the mechanism for this is unclear and not articulated.  

Materials and method  

This section needs to be simplified by realigning it to the study activities (as per comment above). The paper does not address who initiated the idea of implementing digital SP in the postvention space - has this come from the research team, from the sector, the stakeholders? Again, this points to the study sitting more comfortably with implementation science than co-design. It sounds like step 1 was about convincing the stakeholders that digital SP is a good idea and then eliciting their engagement for undertaking future implementation.  

The co-design workshop sounds more like a challenge workshop - where solutions to identified challenges were identified. However, the scope of this discussion was not limited to DSP but rather took the postvention service space more broadly as its focus. I think this, and the subsequent results section, needs some critical reflection by the authors as its key to how the authors position the paper and its contribution to the literature.

It would be useful to the reader if Figure 1 were at the beginning of the section to provide an overview of the steps in the process.

Results

No results are described for step 1 - a summary would be useful. 

10 topics are identified in the text - but Table 1 only shows 8 and repeats 6 of the 8. Some tidying up is warranted here. 

On page 5, after a brief report of topics discussed, the authors state 'In summary, delegates agreed that DSP could support overcoming the stigma of suicide by providing sensitive information, a sense of connectivity and a safe, knowledge-based support that is person-centred'. But this is not clearly elaborated in terms of how. Will the link worker (a social worker?) be sitting down with the person and providing that information? How will a sense of connectivity be achieved? Is the care 'person-centred' because the link worker connects to relevant services based on need? Much more detailed findings here would improve the contribution of the paper. 

The short descriptive paragraphs for each of the themes generally move beyond the contribution that DSP can make in the postvention space - and so perhaps the authors need to think about the contribution of the paper and how to accurately frame these findings. Or develop the findings specifically on DSP in more depth. 

I found Figure 2 confusing and unclear - I think it needs further detailed discussion in the text. 

Discussion

The first paragraph reads like an introduction - rather than the discussion of the findings. Perhaps rework this into the introduction and begin this section with the key findings.  

Conclusion

The short conclusion is a bit weak with statements about 'has the potential' and 'could be introduced'. Perhaps consider the findings in more detail re what they offer the literature in this space. 

Reviewer 2 Report

It is actually very interesting to read a paper that reports on the development and implementation of a suicide bereavement support program. Given that few such papers have ever been published, it is potentially valuable especially for those working in policy and program development and evaluation in this field. Irrespective of my enthusiasm, there are many unclarities that must be addressed to improve the readability of the paper. I detail my concerns below, following the structure of the paper.

Abstract

Line 23: …improve the evidence base: of what?

Line 32: replace ‘significantly’ with substantially or other word (avoid confusion with statistically significance).

Introduction

Line 40: Add a definition of social prescribing here. I know that there was a definition further in the text (line 74). However, few people in this field will be familiar with the concept. Best to explain it immediately.

Line 75: what is a link worker?

Materials and methods

Line 91: what is MRC? Write in full the first time any abbreviation appears in the text.

This whole section is very long. Consider inserting subtitles, for example at line 102 (Step one), line 117 (step two), line 123 (step 3), line 152 (Step 4), and modify each first sentence accordingly. (Do not repeat ‘step one’ etc in each first sentence.

Line 115: The one-day workshop mentioned in this sentence, is that the co-design workshop mentioned in step 3? If so, refer to step 3. If not clarify.

Line 121: consider to include the link to the freely available tool.

Lines 132-133: Here the text refers to people affected by suicide. Earlier it was people bereaved by suicide. Please be consistent.

Line 156: Figure 1 is not readable. The blocks are too dark.

Results

Line 161: Please include the flow of participants from step 1 to step 2 to step 3. How many different individual participants were there in total? Please rephrase paragraph for clarity.

Lines 169-171: This is not a result, but a repetition of the method.

Lines 173-177: Here the text lists 10 topics. However, further in the text (pages 5-6) and in the table there are only 8 topics. Please be consistent.

Line 182: Table 1: i) Seven items are repeated in the table. ii) The wording of the items and the number of items do not match with the items listed in lines 173-177. iii) The content of the table is repeated in the text below the table. As such, the table adds little to the paper. I suggest to delete it completely.

Lines 184-234: Add a subtitle per item, use the name of the items as subtitles.

There is overlap between the description of the items. For example, the items on lines 211 and 234 address timing. There seems to be more overlap between items. I strongly suggest that authors re-read this whole section in order to decide how many items there are, and to delete all repetitions.

Line 226: What is DSP?

Line 244: Here, SD1 is called a postvention service. On line 7, it was a form.

Line 260: The expert group, is that the same group as on line 161?

Discussion

Line 273: Something wrong with the sentence?

Line 304: Limitations

Please add that the ethical aspects regarding applying DSP in the field of suicide bereavement support were not addressed. In fact, this is the biggest omission from this manuscript, and if possible, it would be worthwhile to add a paragraph regarding the ethical issues and challenges.

Line 320: replace ‘significant’ with ‘substantial’ or something similar.

References

Line 405, ref 28: please complete the reference

Line 412: ref 31: at least the name of the journal is missing.

Please re-read your manuscript through the eyes of people such as clinicians or researchers who are involved in postvention, but not necessarily in policy or program development. You want them to understand it as well. Good luck with the revision!

Round 2

Reviewer 1 Report

The authors are to be commended on their engagement with reviewer feedback and their additional work to strengthen the manuscript. The result is a much clearer and useful contribution to the literature. I look forward to reading how this work progresses into implementation.

Author Response

 Response 1: Thank you for your support to improve the manuscript.

Reviewer 2 Report

Thank you for considering my comments. The manuscript has improved a lot. My remaining comment concerns the paragraph on 'ethical challenges'. Surely you cannot include this in the results section, as this was not an aim of the study. It should be a included in the discussion section. 

Otherwise, well done. 

Author Response

Response 1: Thank you for your support to improve the manuscript. We agree and we have moved the discussion of ethical considerations to the discussion section, as suggested. The revised manuscript has been proof read and minor grammatical/typo issues have been resolved.